# PEFTDebias : Capturing debiasing information using PEFTs

**Sumit Agarwal** *    **Aditya Srikanth Veerubhotla** *    **Srijan Bansal** *

Language Technologies Institute, Carnegie Mellon University, Pittsburgh, PA

{sumita, adityasv, srijanb}@andrew.cmu.edu

## Abstract

The increasing use of foundation models highlights the urgent need to address and eliminate implicit biases present in them that arise during pre-training. In this paper, we introduce PEFTDebias, a novel approach that employs parameter-efficient fine-tuning (PEFT) to mitigate the biases within foundation models. PEFTDebias consists of two main phases: an upstream phase for acquiring debiasing parameters along a specific bias axis, and a downstream phase where these parameters are incorporated into the model and frozen during the fine-tuning process. By evaluating on four datasets across two bias axes namely gender and race, we find that downstream biases can be effectively reduced with PEFTs. In addition, we show that these parameters possess axis-specific debiasing characteristics, enabling their effective transferability in mitigating biases in various downstream tasks. To ensure reproducibility, we release the code to do our experiments[1].

## 1 Introduction

In recent years, it has become evident that foundation models such as BERT or GPT-3 (Devlin et al., 2019; Brown et al., 2020) are susceptible to a range of stereotypical societal biases (Jentzsch and Turan, 2022) such as sexism (*gender*) (Kurita et al., 2019) and racism (*race*) (Ahn and Oh, 2021), that are present in the training data. Such bias axes can lead to unfair or discriminatory outcomes (Webster et al., 2021; Barikeri et al., 2021) in various socio-technical scenarios.

Recent research (Ladhak et al., 2023) suggests that biases acquired during pre-training can propagate to downstream models, resulting in superficial text dependencies and potential implicit bias, and a

---
*Equal contribution, names ordered randomly.
[1]https://github.com/sumit-agrwl/peft-debias

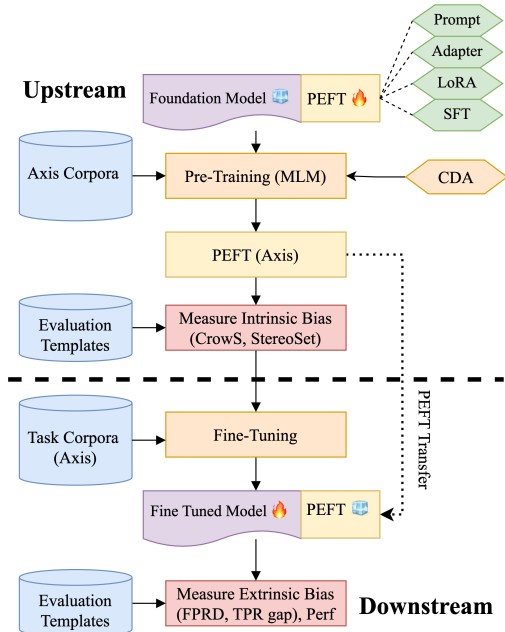

Figure 1: The figure illustrates our proposed PEFTDebias method to debias the fine-tuned model, which consists of two main phases - *upstream* phase where debiasing parameters are acquired through CDA-based PEFT training on axis corpora, evaluated using intrinsic metrics, *downstream* phase, where the debiased PEFT is injected into a trainable model and kept frozen during the fine-tuning process on a task corpora. Bias is measured using extrinsic metrics along the same axis.

higher likelihood of subsequent harmful effects, a concept known as *bias transfer hypothesis* (Bolukbasi et al., 2016; Caliskan et al., 2017). However, most approaches for bias mitigation are primarily applied during fine-tuning to reduce bias in specific downstream tasks or datasets (Park et al., 2018; Zhang et al., 2018). It involves incorporating auxiliary training objectives (Jin et al., 2021), annotation of bias attributes (Liang et al., 2020) and task-specific fairness metrics (Zhang et al., 2020), which poses a challenge for the expanding community of fine-tuning language models.

Previous studies have attempted to address this

issue by first debiasing the model and then fine-tuning it for a specific task. This process referred to as *upstream debiasing* by Jin et al. (2021), and entails fine-tuning the model on upstream tasks while incorporating bias-attribute annotations for debiasing. Subsequently, the model is fine-tuned for the target *downstream* task. Nevertheless, this approach possesses certain limitations: (i) it requires annotated bias attributes for the upstream task as well as supervised data for both tasks and (ii) there is no guarantee that the model will exhibit reduced bias in the downstream task (Steed et al., 2022). This uncertainty arises due to the fact that modifying all parameters of the debiased upstream model might result in the loss of debiased representations. This phenomenon is commonly referred to as *fairness forgetting* (Lauscher et al., 2021).

Inspired by the promising outcomes of PEFT methods, which effectively capture debias information and yield competitive results compared to full model-tuning (Kumar et al., 2023; Lauscher et al., 2021), we hypothesize that employing PEFTs for debiasing on an upstream bias axis could be a viable approach to mitigate bias in a foundation model for any downstream task on the same bias axis. To address this, we present a novel method called PEFTDebias. This approach utilizes PEFTs to capture debiasing information by training the model on axis-specific data during the upstream stage. Subsequently, in the downstream task, the model is fine-tuned while keeping the PEFTs frozen, thereby preserving the upstream debiasing information along that axis. Our contribution can be summarized as:

- We explore the efficacy of training PEFT parameters along a specific bias axis by utilizing axis-based data to transfer bias information to downstream tasks aligned with that axis.

- We evaluate the effectiveness of various PEFT methods in mitigating social biases to determine whether certain PEFT techniques are more efficient than others.

- We examine the transfer capabilities of PEFTs across different datasets to mitigate social biases along specific axes.

## 2 Related Work

Several debiasing methods have been proposed in conjunction with the downstream task, including counterfactual data augmentation (Zmigrod et al.,

2019), dropout regularization (Webster et al., 2020), null-space projection (Ravfogel et al., 2020), adversarial training (Liu et al., 2020), contrastive learning (He et al., 2022). However, these techniques necessitate expensive additional annotation, such as the inclusion of protected attributes, along with the task data. Conversely, (Jin et al., 2021) demonstrate debiasing using only task data, showing its potential for improving generalization. In contrast, (Steed et al., 2022) indicate that debiasing a language model (LM) prior to fine-tuning does not guarantee unbiasedness in the resulting fine-tuned model. Jin et al. (2021) investigate the transferability of debiasing techniques. They begin by applying bias mitigation to a pre-trained model through fine-tuning and subsequently employ it for downstream fine-tuning.

Lauscher et al. (2021); Kumar et al. (2023) show that PEFT methods like Adapters (Houlsby et al., 2019), can be used to debias language models (LMs) while keeping the LM backbone frozen. Hauzenberger et al. (2023) present a method to do debiasining by identifying sparse subnetworks that correspond to different bias axes, which can subsequently be composed. A notable advantage of these approaches is the reduced computational cost and environmental impact associated with debiasing LMs (Hessenthaler et al., 2022). Additionally, it holds the potential for preventing catastrophic forgetting of pre-trained knowledge caused by fine-tuning (Kirkpatrick et al., 2017). However, these techniques are typically applied during the downstream phase and possess the limitations discussed earlier.

## 3 Bias Factors and Datasets

We validate our hypothesis by conducting validation on two widely recognized factors of social bias: *gender stereotyping* and *racial identifiers*. To address occupation-based gender stereotypes, we utilize the BiasBios dataset (De-Arteaga et al., 2019). For the bias related to race, we address the issue of elevated occurrences of false positive outcomes in hate speech predictions using GHC (Kennedy et al., 2018). To show our generalizibility of capturing debiasing information along a specific axis using PEFTs, we show transfer to datasets MNLI (multi genre NLI) (Williams et al., 2018) and LHC (large hate corpus) (Toraman et al., 2022) along gender and race axis respectively.

In order to assess the effectiveness of our debi-

asing techniques in mitigating gender and racial biases, we utilize two intrinsic bias benchmarks, namely CrowS-Pairs (Nangia et al., 2020) and StereoSet (Nadeem et al., 2021), during the initial phase of our evaluation, referred to as the *upstream* stage. StereoSet evaluates a language model's stereotypical associations by employing fill-in-the-blank problems with intra-sentence examples across different bias categories. CrowS-Pairs is an intra-sentence dataset of minimal pairs that compares the language model's masked token probabilities of sentences with disadvantaged or advantaged races fulfilling or violating stereotypes.

In the subsequent *downstream* stage, we evaluate the performance gap of PEFTs across different protected attributes within the specific domain using extrinsic bias metrics. To measure gender bias, we adopt the method proposed by De-Arteaga et al. (2019) to calculate the gender gap in the True Positive Rate (TPR) for each occupation (TPR-GAP). To assess racial bias, we compute the False Positive Rate Difference (FPRD) by comparing the FPR of examples mentioning protected racial attributes to the overall FPR. We calculate FPRD for both the in-domain data and the Identity Phrase Templates Test Sets (IPTTS) (Zhang et al., 2020), which consist of 77k instances. These instances comprise hate and non-hate sentences that mention 25 racial identifiers and are generated using predefined templates. To measure transferability, we evaluate MNLI using FN (fraction of neutrals) in Bias-NLI (Dev et al., 2019), a NLI dataset to measure gender bias, and LHC using IPTTS.

## 4 Methodology

Kumar et al. (2023) demonstrates that incorporating adapters in debiasing during the finetuning process helps. However, transferring adapters between different datasets/tasks is not feasible due to the need to learn data-specific modules. While Lauscher et al. (2021) indicate that learning adapters in the upstream phase contributes to better results during downstream fine-tuning. We propose a novel approach called PEFTDebias which combines elements from both aforementioned methods. It consists of two main phases: the *upstream* phase, responsible for selecting debiasing parameters through PEFTs, and the *downstream* phase, which employs the debiased PEFTs for task debiasing during fine-tuning, as illustrated in Figure 1 and outlined in pseudo-code A.3. We investigate

the viability of multiple PEFTs, including Adapters (Pfeiffer et al., 2021), Prompt Tuning (Lester et al., 2021), LoRA (Hu et al., 2021), and Sparse Fine-tuning (Ansell et al., 2022) (refer A.2).

### 4.1 Upstream Phase

Counterfactual Data Augmentation (CDA) (Zmigrod et al., 2019) is a data-based debiasing technique that swaps attribute words pertaining to a bias (e.g, he/she for binary gender). Parameter efficient debiasing with Adapters (Lauscher et al., 2021) has demonstrated the effectiveness of using CDA to capture debiasing information while minimizing the number of parameters. Consequently, our study aims to explore the application of CDA using PEFT methods for obtaining debiasing parameters. Specifically, we utilize a PEFT to perform CDA on axis-specific data. We extract attribute words from a particular axis and apply them through CDA to obtain debiasing PEFT parameters. Our hypothesis posits that these parameters will proficiently capture task-agnostic debiasing information that is specific to the designated axis.

### 4.2 Downstream Phase

To enable the transferability of debiasing PEFT parameters across datasets, we propose learning debiasing parameters during the upstream phase and injecting them into a trainable language model while keeping PEFT parameters frozen during downstream task fine-tuning. Our hypothesis is that this set of frozen parameters will retain the upstream debiasing effect and safeguard the model against acquiring biases during task finetuning. Consequently, it effectively mitigates biases along the specific axis in the finetuned model.

## 5 Results

Our experimental setup is described in A.4. We present three sets of results: evaluation of the upstream and downstream phases on the same datasets, and the transferability to other datasets.

### 5.1 Upstream Phase

In Table 1, we present the results of our experiments in the upstream setting. The results clearly indicate that the **utilization of PEFTs with CDA not only enhances the performance of LM, but also diminishes intrinsic bias**. Remarkably, both the Prompt Tuning and Adapter techniques demonstrate substantial debiasing effectiveness while either preserving or even enhancing the LM score

when compared to other techniques. For BiasBios, Prompt Tuning shows the highest performance in bias intrinsic scores of CrowS and StereoSet.

| PEFT | SS LM ↑ | SS Score ↓ | CrowS ↓ |
|---|---|---|---|
| **BiasBios** | | **Eval** : Gender | |
| BERT | 85.68 | 60.03 | 57.25 |
| + Full-Debias | 85.74 | 60.28 | 54.96 |
| + Adapter | **86.45** | 57.1 | 53.82 |
| + Prompt | 85.54 | **56.64** | **51.91** |
| + LoRa | 86.21 | 58.85 | 54.20 |
| + SFT | 86.22 | 57.9 | 55.34 |
| **GHC** | | **Eval** : Race | |
| BERT | 83.88 | 57.06 | 62.33 |
| + Full-Debias | 84.01 | **57.03** | **45.63** |
| + Adapter | **85.88** | 58.56 | 55.15 |
| + Prompt | 85.73 | 58.78 | 52.62 |
| + LoRa | 84.89 | 58.20 | 56.12 |
| + SFT | 85.42 | 58.91 | 54.76 |

Table 1: Results in the Upstream setting using BERT as the LM and CDA for performing Debiasing.

## 5.2 Downstream Phase

The results of the downstream experiments are presented in Table 2 where the dataset used in the upstream phase is same as the one in the downstream phase, demonstrating that the PEFTs attain comparable task performance to the BERT baseline (within a 5% margin) with a significant improvement in extrinsic bias metric. This observation suggests that it is possible to **achieve efficient debiasing without significant performance loss. Among the PEFTs, Prompt Tuning stands out for its superior ability to reduce bias**. This finding implies that Prompt Tuning effectively debiases the model in the upstream phase while maintaining its task performance, possibly due to minimal modifications inside the language model (Ding et al., 2022) during forward pass as compared to other PEFTs. Additionally, both BiasBios and GHC exhibit a positive correlation between upstream debiasing performance and downstream bias reduction. This correlation indicates that upstream debiasing can effectively transfer to downstream tasks using PEFTs, facilitating bias mitigation across similar axes. We also study in detail the reduction in bias in BiasBios dataset in A.5

## 5.3 PEFT Transfer

To evaluate the task-agnostic nature of the learned upstream debiasing parameters along a specific axis, we conduct experiments where we apply these

| PEFT | BiasBios (Gender) | | GHC (Race) | | |
|---|---|---|---|---|---|
| | ACC ↑ | TPR-GAP ↓ | F1 ↑ | FPRD ↓ | FPRD$_{IPTTS}$ ↓ |
| FT | 81.29 | 13.05 | **68.76** | 1.01 | **0.01** |
| Full-Debias | 81.27 | 12.86 | 62.48 | 1.07 | 0.08 |
| Adapter | 81.28 | 13.22 | 67.39 | 0.78 | 0.02 |
| Prompt | 81.10 | **11.98** | 67.15 | **0.54** | **0.01** |
| LoRa | 81.28 | 13.67 | 66.91 | 0.73 | 0.12 |
| SFT | **81.34** | 12.04 | 65.06 | 0.59 | 0.25 |
| PEFT | BiasBios → MNLI | | GHC → LHC | | |
| | ACC ↑ | FN ↑ | F1 ↑ | FPRD ↓ | FPRD$_{IPTTS}$ ↓ |
| FT | **80.52** | 0.02 | 91.06 | 0.34 | 0.03 |
| Full-Debias | 80.13 | 0.02 | **91.63** | 0.32 | **0.00** |
| Adapter | 80.11 | 0.02 | 91.47 | 0.33 | 0.01 |
| Prompt | 80.01 | **0.21** | 91.2 | 0.34 | **0.00** |
| LoRa | 80.3 | 0.02 | 91.18 | 0.32 | 0.01 |
| SFT | 80.25 | 0.01 | **91.63** | **0.31** | 0.01 |

Table 2: Task performance and extrinsic bias matrix results in the downstream setting on the BiasBios (gender) and GHC (race) datasets; *same* as those used during the upstream phase (above) and transfer setting on *different* MNLI (gender) and LHC (race) datasets (below)

parameters during the finetuning process for a corresponding task in the same axis on MNLI and LHC. By comparing these results with the ones reported in Table 2, we observe that the performance of the transferred debiasing parameters is comparable to that of full finetuning (FT). While parameters learned from the same task data exhibit the least bias, as indicated by the FPRD and FPRD$_{IPTTS}$ metrics, Table 2 demonstrates that comparable performance can still be achieved through transfer. Notably, the SFT and Prompt Tuning outperform full finetuning on in-domain FPRD metrics when it comes to transfer which also aligns with our findings from previous experiments. In case of MNLI, the performance remains similar to that of full finetuning while Prompt Tuning showing impressive performance for bias scores calculated using BiasNLI. This indicates that **task-agnostic axis-based patch generated by PEFTs work effectively to debias along the same axis across different datasets**.

## 6 Conclusion & Future Work

This research paper introduces PEFTDebias, a novel debiasing approach that utilizes PEFTs to mitigate the biases. PEFTDebias involves two phases: an upstream phase for learning debiasing PEFTs along specific bias axes, and a downstream phase where these PEFTs are incorporated into the model and kept frozen while fine-tuning. Experimental results highlight the effectiveness of Prompt

Tuning for downstream debiasing and the transferability of axis-specific debiasing parameters in mitigating biases across different tasks. Future work includes extending our technique for generative models and tasks, as well as exploring the composition of multiple bias axes (Jin et al., 2021) to address various biases in datasets.

## 7 Limitation

Our research specifically targeted the debiasing of BERT, a widely used language model, and did not encompass other foundational language models such as GPT-3 limiting its scope to the specific context of BERT and its associated biases. We demonstrated the effectiveness of our debiasing techniques on downstream classification tasks. However, it is important to note that these findings may not directly translate to generative language models, as they approach every task as a generation problem. To extend the applicability of our approaches to the broader landscape of all foundational language models, further analysis and investigation would be necessary. We focus our study on mitigating the biases within the dataset, and do not focus on the biases in the annotation of the task labels.

## 8 Ethical Considerations

In this research, we employed a binary gender definition while examining gender bias in pre-trained language models. However, we acknowledge that gender is non-binary and recognize the importance of using a more flexible definition in future studies on gender bias drawing inspiration from previous research (Dinan et al., 2020). Likewise, our investigation of racial bias is limited to a specific set of biased attribute words, representing a narrow definition. It is important to note that we did not explore the potential reduction in harm through the implementation of our debiasing techniques in real-world scenarios. Furthermore, we want to emphasize that all the intrinsic bias benchmarks used in this study possess only positive predictive power. This means that they can identify biased models but cannot confirm a model as unbiased. For instance, a stereotype score of 50% on StereoSet or CrowS-Pairs does not necessarily indicate an unbiased model. The extrinsic measures also rely on few words or templates and cannot comprehensively capture all the stereotypical variations used by humans, Due to these considerations, we urge readers to refrain

from making definitive claims about the debiasing techniques outlined in this paper or applying them directly in real-world settings.

## 9 Acknowledgement

We thank Professors Emma Strubell and Maarten Sap for their valuable guidance and feedback on this work.

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

# A Appendix

## A.1 Bias Axes & Attribute Words

We describe the bias axes and attribute words that we will use in our studies. We mention two different biases, gender and race. Hereby, we present a list of some attribute word examples as well along with the biases.

**Gender** (actor, actress), (boy, girl), (brother, sister), (he, she)

**Race** (black, caucasian, asian), (african, caucasian, asian), (black, white, asian)

## A.2 Paramter Efficient Fine-Tuning (PEFT)

We explore the use of multiple PEFTs, **Adapters**: (Pfeiffer et al., 2021) which are task-specific modules inserted between transformer layers, **Prompt Tuning** : (Lester et al., 2021) which involves incorporating task-specific vectors (prompts) into the input sequence, **LoRA** : (Hu et al., 2021) which integrates trainable low-rank matrices into transformer layers in order to approximate weight updates, and **Sparse Fine Tuning** : (Ansell et al., 2022) builds upon the Lottery Ticket Hypothesis (LTH) to select a sparse sub-network based on the parameters that undergo the most significant changes.

## A.3 Algorithm

---

**Algorithm 1** PEFTDebias training algorithm

---

**Require:** $D_u = \{x_i\}_{i=1}^N$ // unlabelled
**Require:** $D_l = \{(x_i, y_i) \sim P(X,Y)\}_{j=1}^N$ // labelled
   Initialize $\theta_{FM}$
   Initialize $\phi_{PEFT}$

   /* Upstream stage */
   $\phi_{PEFT}^A* \leftarrow Debias(\theta_{FM}, \phi_{PEFT}, D_u, A)$

   /* Downstream stage */
   $\theta_{FM}^* \leftarrow FT(\theta_{FM}, \phi_{PEFT}^A{}^*, D_l)$

   **return** $\theta_{FM}^* \cup \phi_{PEFT}^A{}^*$

---

Our algorithm for debiasing is described in 1. Our method requires an unlabeled in-domain corpus $D_u$ for upstream debasing and a labeled corpus $D_l$ for task-specific fine-tuning in the downstream phase. We use a pretrained foundation model $\theta_{FM}$, and a set of PEFT parameters $\phi_{PEFT}$ which will be used for debiasing the model. In the upstream stage, the backbone model is kept frozen and domain and axis-specific PEFT parameters $\phi_{PEFT}^A{}^*$ for the axis $A$ are obtained. These are then used to finetune the foundation model on the downstream

task while keeping the PEFT frozen to obtain $\theta_{FM}^*$. The final debiased task-specific model is the union of the axis-specific PEFT and the foundation model ($\theta_{FM}^* \cup \phi_{PEFT}^*$)

## A.4 Experimental Setup

We used pre-trained BERT (Devlin et al., 2018) as the starting point for all of our models. We also applied text normalization to GHC datasets to remove URLs and user mentions using tweet based processing [2]. For the upstream experiments, we trained our models with MLM and CDA on the BiasBios dataset and the other datasets using a learning rate of $1e^{-5}$ and a batch size of 128 and 32 respectively. We ran MLM for 10,000 steps and evaluated the models every 1,000 steps. We selected the models with the lowest loss for our experiments. For the downstream experiments, we used a batch size of 32 and trained our models for 10 epochs. We ensured that all PEFTs have similar number of parameters, being 1% of the base LM, to keep them comparable. For the downstream experiments, we used a batch size of 32 and trained our models for 10 epochs. We chose the models with the best task metrics for analysis. For GHC and Stormfront datasets, which had few hateful examples compared to non-hateful ones, we weighted the loss of hateful examples by a factor of 10 for GHC and 6.7 for Stormfront, based on their proportions in the data. We compared our methods with two baselines: BERT in the pre-trained setting and BERT in the fine-tuned setting (Full-Debias). Our implementation is based on the AdapterHub [3].

## A.5 Reduction in bias

We conducted a comparison of the TPR-GAP performance of CDA debiasing techniques using FT and Prompt Tuning on the BiasBios dataset (see Figure 2, specifically focusing on occupations categorized as male and female. Our findings indicate that debiasing with Prompt Tuning yields better results compared to FT, as evidenced by a decrease in the TPR for gender-dominant professions. We observed that certain female-dominated professions such as dietitian and interior designer exhibit reduced correlation with the female gender, while male-dominated professions like surgeon and comedian also demonstrate a decrease in correlation with the male gender. Although we did not observe significant changes in the gap for professions

---

[2]link to script
[3]https://adapterhub.ml/

like rapper and psychologist, we encountered an issue of over-correction, resulting in a reversed gap for poet and accountant. This discrepancy can be attributed to the limited number of examples available for these particular professions.

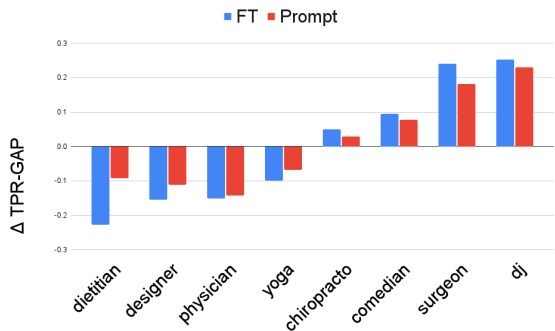

Figure 2: Comparing the TPR-GAP performance of CDA debiasing using FT and Prompt Tuning on the Biasbios dataset across different occupations.