# OpenReview forum: "PEFTDebias : Capturing debiasing information using PEFTs"
_EMNLP/2023/Conference — EMNLP 2023 Main_

### Official Review · Reviewer_xV67 · 2023-08-04

**Soundness:** 4

**Excitement:**

4: Strong: This paper deepens the understanding of some phenomenon or lowers the barriers to an existing research direction.

**Missing References:**

Modular and On-demand Bias Mitigation with Attribute-Removal Subnetworks
Lukas Hauzenberger, Shahed Masoudian, Deepak Kumar, Markus Schedl, Navid Rekabsaz
In Findings of the Association for Computational Linguistics: ACL (Findings of ACL), 2023

**Paper Topic And Main Contributions:**

The work explores the important topic of exploiting upstream debiasing in downstream tasks, and proposes leveraging parameter-efficient fine tuning (PEFT) methods with a specific downstream training strategy. The upstream bias mitigation is done using CDA, learned through various PEFT architectures appended to BERT, and evaluated on BiasBIOs, GHC, as well as MNLI and LHC. The results show that upstream bias mitigation when freezed during downstream training can transform its effectivity.

**Questions For The Authors:**

No question.

**Reasons To Accept:**

- Well written paper with a clear and to-the-point objective which perfectly fits to the scope of a short paper.
- The introduced approach, while relatively simple, has practical benefits, particularly in terms of exploiting LLM debiasing techniques in downstream tasks.
- Extensive experiments in terms of various PEFT approaches.

**Reasons To Reject:**

- The method is only evaluated on BERT and could have benefited from evaluation on more LLMs.

**Reproducibility:**

4: Could mostly reproduce the results, but there may be some variation because of sample variance or minor variations in their interpretation of the protocol or method.

**Reviewer Confidence:**

4: Quite sure. I tried to check the important points carefully. It's unlikely, though conceivable, that I missed something that should affect my ratings.

---

> ### Author Rebuttal · Authors · 2023-08-29
>
> Thank you for appreciating our work. We would like to provide clarifications for some of your concerns and questions.
>
> > The method is only evaluated on BERT and could have benefited from evaluation on more LLMs.
>
> This is the same point as question 1 from Reviewer 2, and is addressed in our response there.
>
> > Missing References:
> Modular and On-demand Bias Mitigation with Attribute-Removal Subnetworks Lukas Hauzenberger, Shahed Masoudian, Deepak Kumar, Markus Schedl, Navid Rekabsaz In Findings of the Association for Computational Linguistics: ACL (Findings of ACL), 2023
>
> Thank you very much for pointing out this work, we will surely include this in the camera-ready draft of our paper.

---

### Official Review · Reviewer_A5DV · 2023-08-05

**Soundness:** 4

**Excitement:**

4: Strong: This paper deepens the understanding of some phenomenon or lowers the barriers to an existing research direction.

**Paper Topic And Main Contributions:**

This work focus on debiasing pre-trained models, and proposes a two-stage approach called PEFTDebias, which training PEFT parameters along a specific bias axis while keeping base model frozen in the upstream task, then fine-tune the base model on downstream tasks while keeping the PEFT parameters frozen. Experimental results on several PEFT methods and different datasets show that the proposed methods effectively reduce the downstream biases and have transferability in various datasets.

**Reasons To Accept:**

1. This work focus on an important and foundation task that try to mitigate biases in pre-trained models.
2. Clear motivation and comprehensive experiments, the experiments provide a sufficient support for its claims.


**Reasons To Reject:**

The work mainly conducts experiments on encoder-based model (i.e., Bert), but has no idea of generative models, like GPT.

**Reproducibility:**

4: Could mostly reproduce the results, but there may be some variation because of sample variance or minor variations in their interpretation of the protocol or method.

**Reviewer Confidence:**

4: Quite sure. I tried to check the important points carefully. It's unlikely, though conceivable, that I missed something that should affect my ratings.

---

> ### Author Rebuttal · Authors · 2023-08-29
>
> Thank you for taking the time to review our work. To address your question, this works aims to present a preliminary study on evaluating the role of PEFT based techniques for debiasing LMs. While this study only shows results on Encoder-only models, our approach can be easily extended to other architectures such as Decoder-only and Encoder-Decoder architectures. We will leave these explorations to future work.

---

### Official Review · Reviewer_LNGT · 2023-08-11

**Soundness:** 3

**Excitement:**

3: Ambivalent: It has merits (e.g., it reports state-of-the-art results, the idea is nice), but there are key weaknesses (e.g., it describes incremental work), and it can significantly benefit from another round of revision. However, I won't object to accepting it if my co-reviewers champion it.

**Justification For Ethical Concerns:**

-

**Missing References:**

-

**Paper Topic And Main Contributions:**

The paper's authors suggest an effective approach for fine-tuning, specifically a parameter-efficient fine-tuning method, to mitigate bias issues in trained models. They make use of axis-based data to harness information relevant to downstream tasks that align with these axes. This approach consists of two primary phases. Initially, during the upstream phase, they employ a parameter-efficient fine-tuning method to identify debiasing parameters. Subsequently, in the downstream phase, the method employs these debiased parameters for fine-tuning. The authors demonstrate the effectiveness of their proposed algorithm through evaluations in various social bias tasks, highlighting its superiority.

**Questions For The Authors:**

-

**Reasons To Accept:**

The proposed method not only improves the performance of the language model but also effectively reduces inherent bias. This indicates that the method possesses the capability to enhance both debiasing and fundamental performance aspects.

**Reasons To Reject:**

Comprehending the operation of the proposed method is challenging, even though the paper is concise. Utilizing pseudo-code could potentially offer clearer insight into the workings of both the upstream and downstream phases, as compared to relying solely on Figure 1.

Additionally, despite the inclusion of "Parameter efficient" in the name of the proposed method, there appears to be a lack of performance evaluation specifically addressing its efficiency. To enhance the comprehension of the proposed method's effectiveness, it would be beneficial to incorporate an analysis that delves into the computational costs associated with the approach.

**Reproducibility:**

2: Would be hard pressed to reproduce the results. The contribution depends on data that are simply not available outside the author's institution or consortium; not enough details are provided.

**Reviewer Confidence:**

3: Pretty sure, but there's a chance I missed something. Although I have a good feel for this area in general, I did not carefully check the paper's details, e.g., the math, experimental design, or novelty.

**Typos Grammar Style And Presentation Improvements:**

-

---

> ### Author Rebuttal · Authors · 2023-08-29
>
> Thank you for 	providing your valuable feedback. We would like to provide some clarifications for our approach.
>
> > Comprehending the operation of the proposed method is challenging, even though the paper is concise.
>
> In the final version of our draft, we will add a detailed pseudocode for our approach in the appendix, and review the wording of our method to make it more clear to understand.
>
> > Additionally, despite the inclusion of "Parameter efficient" in the name of the proposed method, there appears to be a lack of performance evaluation specifically addressing its efficiency. To enhance the comprehension of the proposed method's effectiveness, it would be beneficial to incorporate an analysis that delves into the computational costs associated with the approach.
>
> 1. Through our experimentation we find that PEFTs help capture debiasing information along a particular axis. The efficiency of our approach lies in the limited number of parameters (1% of the total parameters) utilized in the Upstream debiasing stage, which is a one-off experiment within a specific domain.
> 2. The downstream phase operates akin to full fine-tuning, with the PEFT parameters being frozen. This approach results in comparable computational time (within 2%) during the downstream phase on BiasBios, the largest dataset for our experiments. Our observations indicate that training times are similar. Additional details regarding computational analysis will be included in the appendix of the final camera-ready draft.
> 3. By demonstrating transfers to various datasets, we also indicate that these debiasing parameters could be efficiently applied across diverse datasets without needing to retrain the parameters. This not only saves a significant amount of time but also benefits when applying them to multiple datasets simultaneously.
>
> > Would be hard pressed to reproduce the results. The contribution depends on data that are simply not available outside the author's institution or consortium; not enough details are provided.
>
> To ensure the reproducibility of our results, we will share the repository that was used to generate the results presented in the final camera-ready version of our paper.

---

### Meta-Review · Area_Chair_VTF8 · 2023-09-20

**Recommendation:** 4

**Metareview:**

In their work “PEFTDebias : Capturing debiasing information using PEFTs“, the authors propose to use parameter-efficient fine-tuning (PEFT) for two-stage debiasing of language models. They evaluate several PEFT variants on BERT and four datasets across two bias dimensions (gender, race), and demonstrate that their methodology is effective for reducing biases.

Overall, the reviewers appreciate this work (soundness, excitement). Given that this is a shortpaper submission, the scope is generally feasible. However, they also note that the claims would be stronger with results on more/ other base models and that more emphasis on the efficiency aspect (as suggested by the title of this work) would enrich the discussion.

---

### Decision · Program_Chairs · 2023-10-07

**Decision:**

Accept-Main

**Comment:**

In their work “PEFTDebias : Capturing debiasing information using PEFTs“, the authors propose to use parameter-efficient fine-tuning (PEFT) for two-stage debiasing of language models. They evaluate several PEFT variants on BERT and four datasets across two bias dimensions (gender, race), and demonstrate that their methodology is effective for reducing biases.

Overall, the reviewers appreciate this work (soundness, excitement). Given that this is a shortpaper submission, the scope is generally feasible. However, they also note that the claims would be stronger with results on more/ other base models and that more emphasis on the efficiency aspect (as suggested by the title of this work) would enrich the discussion.